# Evaluating the Mode of Antifungal Action of Heat-Stable Antifungal Factor (HSAF) in *Neurospora crassa*

**DOI:** 10.3390/jof8030252

**Published:** 2022-03-01

**Authors:** Xiaodong Liu, Xianzhang Jiang, Haowen Sun, Jiawen Du, Yuhang Luo, Jianzhong Huang, Lina Qin

**Affiliations:** 1National Joint Engineering Research Center of Industrial Microbiology and Fermentation Technology, College of Life Sciences, Fujian Normal University, Fuzhou 350108, China; xdliu777@163.com (X.L.); jiangxz@fjnu.edu.cn (X.J.); sunhaowen96@163.com (H.S.); 15880015320@163.com (J.D.); 491132253lyh@gmail.com (Y.L.); 2Institute of Biotechnology, Fujian Academy of Agricultural Sciences, Fuzhou 350003, China

**Keywords:** HSAF, antifungal, transcriptome, *Neurospora crassa*, cell wall, autophagy

## Abstract

Heat-stable antifungal factor (HSAF) isolated from *Ly**sobacter enzymogenes* has shown a broad-spectrum of antifungal activities. However, little is known about its mode of action. In this study, we used the model filamentous fungus *Neurospora crassa* to investigate the antifungal mechanism of HSAF. We first used HSAF to treat the *N. crassa* strain at different time points. Spore germination, growth phenotype and differential gene expression analysis were conducted by utilizing global transcriptional profiling combined with genetic and physiological analyses. Our data showed that HSAF could significantly inhibit the germination and aerial hyphae growth of *N. crassa.* RNA-seq analysis showed that a group of genes, associated with cell wall formation and remodeling, were highly activated. Screening of *N. crassa* gene deletion mutants combined with scanning electron microscopic observation revealed that three fungal cell wall integrity-related genes played an important role in the interaction between *N. crassa* and *L. enzymogens.* In addition, Weighted Gene Co-Expression Network Analysis (WGCNA), accompanied by confocal microscopy observation revealed that HSAF could trigger autophagy-mediated degradation and eventually result in cell death in *N. crassa*. The findings of this work provided new insights into the interactions between the predatory *Lysobacter* and its fungal prey.

## 1. Introduction

Heat-stable antifungal factor (HSAF) is an antimycotic compound isolated from a biological control strain *Ly**sobacter enzymogenes*. It belongs to polycyclic tetramate macrolactams (PTMs) and has shown a broad-spectrum of antifungal activities (Figure 1A) [1]. *Lysobacter enzymogenes* is a Gram-negative soil bacterium belonging to the genus *Lysobacter* and the Xanthomonadaceae family. This bacterium has twitching motility and can produce a variety of extracellular enzymes, including chitinase, protease, cellulase and β-1,3-glucanase [2,3,4,5] and produce antibiotic compounds such as β-lactams and HSAF [6,7]. It has an obvious antagonistic effect on many kinds of microorganisms including fungi, oomycetes, nematodes as well as bacteria [8,9,10,11]. In addition to its antagonistic effect, the *L. enzymogenes* also directly attach to the hyphae of filamentous fungi, and subsequently cause pathogenic infection of the fungal host [8]. Therefore, as a new type of biocontrol bacteria, *L. enzymogenes* has great potential for application in the biological control of agricultural diseases and pharmaceutical treatments. Understanding the antifungal mechanism of *L. enzymogenes* is one of the prerequisites for its broad application.

In recent years, reports on the biocontrol mechanism of *L. enzymogenes* mainly focus on its extracellular hydrolase and secondary metabolites. HSAF is one of the secondary metabolites of *L. enzymogenes*. Due to its antifungal effect, HSAF has received extensive attention, mainly focusing on its antifungal activity, biosynthesis mechanism, separation and yield improvement [1,12,13,14,15]. The mechanism of HSAF that causes fungal growth defects has been studied in many fungal species including *Aspergillus nidulans*, *Colletotrichum fructicola*, *Alternaria alternata* (Fries) Keissler and *Candida albicans*. In *A. nidulans,* HSAF has been reported to affect the polarization of mycelia by disrupting ceramide biosynthesis [16]. In *C. fructicola*, HSAF disrupts the coordination between cytokinesis and nuclear division, induces the production of reactive oxygen species in conidia and destroys the integrity of the conidial cell wall. HSAF specifically targets biological processes required for the formation and elongation of the germ tube but does not affect nuclear division [17]. In *A. alternata*, transcriptomics analysis demonstrates that HSAF disrupts the metabolic network, including the adenosine 5′-monophosphate (AMP)-activated protein kinase (AMPK) signaling pathway, sphingolipid metabolism, carbon metabolism, TCA (tricarboxylic acid) cycle, cell cycle, etc., and ultimately leads to apoptosis [15]. In *C. albicans*, HSAF induces apoptosis by promoting the accumulation of reactive oxygen species (ROS), and molecular dynamics (MD) simulation revealed the theoretical binding mode of HSAF to β-tubulin of *C. albicans* [18]. Although a lot of research has been reported to study the mode of action of HSAF, the target of HSAF is still unknown and the cellular biological mechanism of HSAF has still not been fully elucidated.

*Neurospora crassa* is the key organism in the history of genetics, biochemistry and molecular biology in the 20th century. As a model organism, it is most commonly known for its role in proving the “one gene, one enzyme” hypothesis. Over the years, varieties of molecular, genetic and biochemical techniques and avant-garde tools developed for *N. crassa*, in particular, the availability of a full genome deletion strain set made *N. crassa* valuable as a classical genetic model to investigate a myriad of fundamental biological processes [19]. In this study, we evaluated the antifungal effect of HSAF against *N. crassa* and performed RNA-Seq analysis to assess the genome-wide gene expression differences among different time points of the HSAF treatments. Based on transcriptome analysis, we consequently performed a systematic screening of 36 *N. crassa* deletion mutants to identify three genes with negative effects on HSAF resistance. We further used scanning electron microscopy (SEM) to observe the physical interaction between different *L**. enzymogenes* and *N. crassa* mutants. In addition, we conducted WGCNA analysis to reveal that HSAF was able to trigger the autophagy pathway in *N. crassa* and further used confocal microscopy observation to verify this hypothesis. On the basis of our findings, we proposed a novel strategy for *L**. enzymogenes* to infect its host prey.

## 2. Materials and Methods

### 2.1. Strains and Culture Conditions

*L. enzymogenes* strains used in this study include OH11, the wildtype [20] and (-) HSAF, an HSAF-nonproducing mutant with *hsaf-pks* deletion of OH11 [1]. *L. enzymgenes* strains were cultured on 10% TSA (Tryptic Soy Agar) or in 10% TSB (Tryptic Soy Broth) at 28 °C. The *N. crassa* wildtype strain OR74A (FGSC2489) and available single-gene deletion mutants were obtained from the Fungal Genetics Stock Center (http://www.fgsc.net/ accessed on 1 January 2022). *N. crassa* strains were grown on Vogel’s media [21]. For the HSAF sensitivity assay, the wildtype *N. crassa* conidia were harvested with distilled water and inoculated into Vogel’s media plates with indicated concentrations of HSAF as shown in Figure 1. The final concentration of conidia was 1 × 10^6^ conidia/mL. After 36 h of growth at 25 °C in an incubator, the colony sizes were measured, and the plates were used to take a photograph of colony morphology with a digital camera and perform microscopy observation. 

### 2.2. Physical Interactions of L. enzymogenes and N. crassa

*L. enzymogenes* wildtype strain OH11 and mutant (-)HSAF were resuscitated to one Luria broth (LB) plate containing 50 μg/mL Kanamycin, respectively. Two days later, the resuscitated strains were transferred to 10% TSB medium and cultured at 30 °C for 48 h at 200 rpm. The culture was centrifuged at 6000 rpm for 3 min to discard the supernatant media. The precipitated cells were washed with sterile water twice and re-suspended in sterile water with OD_600_ = 1. The prepared *L. enzymogenes* suspension was added into a 6-well plate at 3 mL per well. The conidia of *N. crassa* wildtype and mutant strains were inoculated into cellophane (1 cm in diameter), and placed in Vogel’s solid plate and cultivated overnight, respectively. The cellophane with germinating mycelia was transferred into 6-well plates containing *L. enzymogenes* suspension and incubated at 28 °C for different time points (0 min, 6 h). After a corresponding time of interaction, three cellophanes was immediately removed and fixed with 2% glutaraldehyde. Thirty-six samples were fixed with 5% glutaraldehyde for 4 h, washed with phosphate buffer three times with an interval of 10–15 min, fixed with 1% osmium for 4 h, and washed with distilled water three times with an interval of 10–15 min. Samples were dehydrated in an ascending ethanol series (50%, 70%, 80%, 90%, 100%) for 15 min per ethanol concentration, replaced with 100% ethanol three times, replaced with tert-butanol twice. After freeze drying, the 36 specimens were coated with gold-palladium for 3 min. Scanning electron microscopy (SEM) observation was performed using scanning electron microscope (JSM-6380LV, JEOL, Japan).

### 2.3. RNA-Seq and Data Analysis

*N. crassa* conidia were inoculated into 3 mL of liquid medium at 10^6^ conidia/mL in a deep 24-well plate and grown at 25 °C on a rotary shaker (200 rpm) for 24 h. The cultures were then added with 40 μM HSAF and treated for 0, 5, 30 and 60 min, respectively. For each timepoint, three replicates were set for RNA-seq analysis. The biomass from different time points of HSAF treatment was filtered and frozen in liquid nitrogen for RNA extraction. RNA-seq libraries were prepared using NEBNext Ultra^TM^ RNA Library Prep Kit for Illumina (NEB, Ipswich, MA, USA) and sequenced on the Illumina Genome Analyzer IIx and HiSeq 4000 platforms from Novogene (Beijing, China). Trimmomatic was used to remove adaptor contamination and check the quality of the raw RNA sequencing reads. Differential expression analysis was performed using DESeq2. The resulting *p* values were adjusted using Benjamini and Hochberg’s approach for controlling the false discovery rate. The FDR < 0.01 and Fold Change ≥ 2 were set as the threshold for significantly differential expression. The Venn diagram was generated by BMKCloud (www.biocloud.net, accessed on 1 January 2022). FungiFun2 online resource (https://elbe.hki-jena.de/fungifun/, accessed on 1 January 2022) and KEGG as the classification ontology were used for functional enrichment analysis. Gene annotations are extracted from the fungi database (https://fungidb.org/fungidb/app, accessed on 1 January 2022) or inferred from the homology of characteristic genes of related fungi. The heat map was generated by R software pheatmap package. The “WGCNA” R package (v1.63) [22] was used to construct the co-expression network of the genes correlated with HSAF. 

### 2.4. The Effect of HSAF on the Protoplast Regeneration of N. crassa

The conidia of *N. crassa* were inoculated into Vogel’s liquid medium and cultivated for 4 h. The germinated mycelia were filtered with a 200-mesh sieve and digested by 10 mg/mL lysing enzyme (Sigma, Shanghai, China) and 1 mg/mL cellulase ‘‘ONOZUKA’’ R-10. The generated protoplasts were washed and re-suspended twice with 1 M sorbitol. Then 1 × 10^9^/mL of protoplasts in 1 M sorbitol solution was used to test the effect of HSAF on protoplasts regeneration. Three μL of protoplasts suspension was added into 3 mL of Vogel’s media plus 1 M sorbitol and 30 μM HSAF in a deep 24-well plate, and cultivated at 25 °C on a rotary shaker (200 rpm) for 10 h. As a negative control, protoplasts were re-suspended into distilled water and were set to eliminate the effect of conidia contamination. The experiment was carried out with three replicates. 

### 2.5. Confocal Microscopy Observation of the Structure of Cell Wall and Nuclei

*N.**crassa* conidia were inoculated into Vogel’s media with or without 30 μM HSAF and cultivated at 25 °C on a rotary shaker (200 rpm) for 6 h. The germlings were dropped onto the glass slide and washed with phosphate buffer saline (PBS, pH7.0) three times. The germlings for cell wall observations were stained with 1 mg/mL Calcofluor white (CFW) for 1 min. The germlings for nuclei observation were stained with 100 ng/mL 4′,6-diamidino-2-phenylindole (DAPI) for 5 min. The stained-glass slides were then washed with PBS for another three times. Images were obtained using a laser scanning confocal microscope TCS SP8 with 63 × 2 NA water immersion objective and a 405 nm (CFW) or 405 nm (DAPI) laser.

### 2.6. Confocal Microscopy Observation of the Structure of ER and Golgi Apparatus

The *N. crassa* strain containing GFP-tagged NCA-1 protein was used to observe the structure of ER [23]. The *N. crassa* strain bearing the GFP-tagged VRG-4 protein was used to observe the structure of the Golgi apparatus [24]. Germlings for microscopy were grown on Vogel’s medium supplemented with or without 30 μM HSAF, at 25 ℃ for 6 h. Germlings were observed using a 63 × 2 NA water immersion objective on a Leica TCS SP8 microscope with a 488 nm laser.

## 3. Results

### 3.1. The Growth Inhibitory Effects of N. crassa by HSAF

As a first step toward investigating the antifungal mechanism of HSAF using the model filamentous fungus *N. crassa,* the effect of HSAF on its growth and morphology was tested. *N. crassa* was cultured on Vogel’s agar plates amended with HSAF at different concentrations (0, 20, 30, 40, 50, 60 μM). Colony sizes and morphology were documented at 36 h post-inoculation. We observed that the sizes of the colonies of *N. crassa* were reduced with an increased concentration gradient of HSAF (Figure 1B,C). The spore germination rate was also determined and the result demonstrated that HSAF significantly inhibited the germination of *N.*
*crassa* conidia in a dose-dependent manner (Figure 1D). Growth of *N. crassa* on Vogel’s solid plates containing HSAF showed that low concentration of HSAF (20 μM) caused the conidia of *N. crassa* to germinate abnormally, while the high concentration of HSAF (60 μM) completely inhibited the germination. Microscopic examination showed that the morphology of mycelium is significantly shortened, twisted and hyper-branched under HSAF treatment (Figure 1E). As reported for other crop and clinical fungal pathogens [16,17,18], these data suggested that a certain amount of HSAF could inhibit the germination and polarized growth of *N. crassa,* enabling it to be a reasonable model to investigate the antifungal mechanism of HSAF.

### 3.2. Genes Involved in Cell Wall Formation and Remodeling Were Highly Activated at the Beginning of HSAF Treatment 

To gain insight into the molecular basis of the relationship between HSAF treatment and the growth defects of *N. crassa,* transcriptome profiling was performed to compare global changes in gene expression levels among the *N. crassa* strains treated with HSAF for 0 min, 5 min, 30 min and 60 min. These samples for RNA-seq were accordingly named as HSAF-0, HSAF-5, HSAF-30 and HSAF-60, respectively. Principal-component analysis (PCA) and Euclidean distance analyses of the 12 sets of RNA-seq data showed that the biological replicate samples clustered together and that the expression pattern of HSAF-5 was similar to that of the HSAF-30, but different from that of HSAF-0 and HSAF-60 (Appendix A). The numbers of reads per kilobases of transcript, per million mapped reads (RPKM) for all genes, were calculated for biological replicates, and differential gene expression analyses were investigated (fold change ≥ 2, and *p* ≤ 0.01) between HSAF-5 and HSAF-0, HSAF-30 and HSAF-0, HSAF-60 and HSAF-0, HSAF-30 and HSAF-5 and HSAF-60 and HSAF-30 (Appendix A). Gene Ontology (GO) analysis, which is an international standard gene functional classification system, was then performed for each group of differentially expressed genes (DEGs) (Appendix A).

These analyses showed that a group of genes associated with fungal cell wall organization were significantly upregulated in HSAF-5 and HSAF-30 samples compared to that in HSAF-0 samples, however, the expression levels of most of these genes were significantly decreased in HSAF-60 samples relative to that in HSAF-30 (Figure 2A and Appendix A). Considering that the cell wall is the first barrier of fungi against HSAF and fungal cell walls are commonly used as a target for developing antifungal drugs, which usually act by inhibiting β-D-glucan synthase, chitin synthase and the glycosylphosphatidylinositol (GPI) anchor pathway [25], we therefore focused on investigating whether HSAF could affect the cell wall formation and remodeling. The germ tubes of *N. crassa* strains with or without HSAF treatment were stained with CFW and observed by confocal microscopy. For the first 1 h post-inoculation, microscopic changes were not observed, however, at 3 and 5 h post-inoculation, HSAF-treated germ tubes showed a significantly enhanced fluorescence signal compared to the non-treated control, especially at the region closest to the tip (Figure 2B). These data suggested that HSAF treatment could enhance the synthesis of the cell wall of *N. crassa*. The possible interpretation for this was that the target of HSAF could be the substance that was involved in the maintenance of the fungal cell wall. The interaction between HSAF and its target will ruin cell wall integrity, which thereby triggers the cell to produce more cell wall components to temporarily compensate for HSAF-caused impairment.

### 3.3. N. crassa Mutants Lack of Genes Encoding Chitin Synthases and Cell Wall Glucanase Showed Higher HSAF Resistance

To test the hypothesis that the target for HSAF were cell wall-related materials, protoplasts of *N. crassa* were prepared to check whether HSAF could affect protoplasts’ regeneration (Figure 3A). To avoid the false positive results caused by incomplete cell wall removal during protoplast preparation, protoplasts suspended in aqueous solution were used as a negative control, since protoplasts will burst osmotically in distilled water. The results shown in Figure 3B demonstrated that HSAF treatment could not affect the regeneration of protoplasts, however, the growth after regeneration was still inhibited. This data suggested that the potential target of HSAF could be the components of fungal cell wall.

As a model filamentous fungus, *N. crassa* has a near full genome deletion strain set collection. To investigate the correlation between the fungal cell wall and HSAF resistance and further identify the potential target, 36 available homokaryotic cell wall-related gene deletion mutants were screened to test HSAF susceptibility. Compared to WT, mutants bearing a deletion of the *chs-9* gene (NCU02592), *chs-1* gene (NCU 03611) and *gh16-5* (NCU05686) gene showed significantly higher HSAF resistance (Figure 4A), indicating that these cell wall remodeling enzymes could be conceivable targets of HSAF.

Both *chs-9* and *chs-1* were annotated as chitin synthase-9 and chitin synthase-1 in *N. crassa* based on sequence alignment. The *gh16-5* was annotated as cell wall glucanase, however, the detail function of these genes in chitin synthesis and cell wall formation was not clear. CFW staining was therefore performed to test the effect of deletion of these genes on cell wall structure. The data shown in Figure 4B suggested that all of these genes’ deletion mutants displayed a similar cell wall structure as that in WT. Meanwhile, the effect of these genes in protoplasts’ regeneration was also tested, and the result showed that deletion of each of these genes did not affect protoplasts’ regeneration (data not shown). These data suggested that these cell wall remodeling enzymes did not play the key role in cell wall formation. However, it could be involved in the antimycotic process of HSAF.

### 3.4. The Interaction between HSAF and the Identified Cell Wall Related Enzymes above Played a Key Role in the Degradation of Cell Wall of N. crassa by L. enzymogens

It has been reported that HSAF plays a crucial role in the process of *L. enzymogens* infecting the plant pathogenic oomycete *Pythium aphanidermatum. L. enzymogens* mutants that lost the ability to produce HSAF are unable to adhere, penetrate and degrade the hyphae of *P. aphanidermatum* [10]. To elucidate the molecular mechanism of how *L. enzymogens* infects *N. crassa*, especially how HSAF interacted with the fungal cell wall, we co-cultured different *L. enzymogenes* mutants and *N. crassa* mutants and used scanning electron microscopy (SEM) to examine the change that occurred to the fungal cell wall. When *N. crassa* wildtype strain and *L. enzymogenes* wildtype strain were co-cultured for 5 h, *L. enzymogenes* could use specific proteins such as chitinases and other glycoside hydrolases to punch holes in the surface of *N. crassa*. However, when the *N. crassa* wildtype strain was co-cultured with a *L. enzymogenes* mutant, in which HSAF synthesis was blocked, *L. enzymogenes* was unable to punch holes anymore (Figure 5A), suggesting that HSAF is crucial for *L. enzymogenes* to penetrate the fungal cell. This observation result was consistent with that reported in *P. aphanidermatum* [10].

Considering that the *N. crassa* mutants ∆*chs-1*, ∆*chs-9* and ∆*gh16-5* showed higher HSAF resistance, to test whether a lack of these genes could affect the infection process, we consequently co-cultured each of these three mutants with the *L. enzymogenes* wildtype strain and the HSAF-nonproducing *L. enzymogenes* strain, respectively. SEM results (Figure 5B–D) showed that after 5 h of co-culture, the *L. enzymogenes* wildtype strain could adhere and aggregate onto the hyphae of all of these *N. crassa* mutants, however, no pores in the surface of these *N. crassa* deletion strains were observed. For the HSAF-nonproducing *L. enzymogenes* strain, as expected, it could not make any holes, as it interacted with *N. crassa* wildtype strains. Moreover, we were surprised to find that the number of cells that adhered to the hyphae of these *N. crassa* deletion mutants was significantly decreased compared to the interaction between the *L. enzymogenes* wildtype strain and *N. crassa* deletion mutants. However, this phenomenon was not observed in the physical interaction between the HSAF-nonproducing *L. enzymogenes* strain and *N. crassa* wildtype strain (Figure 5A). These data suggested that HSAF from *L. enzymogenes*, cell wall-related chitin synthase CHS-1 and CHS-9, as well as cell wall glucanase GH16-5 from *N. crassa* could be important factors associated with how *L. enzymogenes* breaks the protective barriers of *N. crassa,* which is the first step of the infection process. The absence of any of these factors would prevent *L. enzymogenes* from making holes in the surface of *N. crassa*, implying that the direct or indirect interaction between HSAF and these identified cell wall-related enzymes played a key role in the degradation of the cell walls of *N. crassa* by *L. enzymogen**es.*

### 3.5. Weighted Gene Correlation Network Analysis (WGCNA) Revealed That HSAF Treatment Caused Autophagy-Mediated Degradation in N. crassa

From the RNA-seq data analysis, a total of 4124 differentially expressed genes (DEGs) were identified when the simple pairwise comparisons between each time-point (5 min, 30 min and 60 min) of HSAF treatment versus standard no-treatment samples were analyzed (*p*-adj < 0.05) (Appendix A). Weighted gene correlation network analysis (WGCNA) was further performed based on the above DEG analysis. Seven gene co-expression modules were generated after similar clusters were merged (Figure 6A). These co-expression modules were partially positively correlated with HSAF treatment and partially negatively correlated. Among them, the module ME11 had the highest degree of correlation in the positive correlation relationship (correlation coefficient = 0.72, *p* value = 0.009) (Figure 6A). There were three genes included in the module ME11 containing *atg3* (NCU01955), *atg5* (NCU04662), *atg12* (NCU10049). All of these three genes belonged to Atg8 ligase activity gene ontology term (GO:0019776) (Figure 6B), which may involve in autophagy degradation through interacting with Atg8. In yeast and mammalian cells, ATG8 (autophagy-related 8) is a distinctive ubiquitin-like protein tightly binding with lipid phosphatidylethanolamine (PE). Atg8-PE recognizes the Atg8-family interacting motif in cargos and promotes autophagosome biogenesis to result in autophagy-mediated degradation [26]. In the process of autophagy, ATG3 acts as an E2 ubiquitin-like conjugating enzyme in the ATG8 binding system, helping to extend the phagocytic cells [27]. The combination of ATG12 and ATG3 regulates mitochondrial homeostasis and cell death [28]. The ATG12-ATG5/ATG16 complex contributes to form the autophagosome membrane and modulates the immune system and crosstalks with apoptosis [29].

To further prove that HSAF treatment triggered the autophagy pathway in *N. crassa*, we then observed the structure of autophagy-related organelles including endoplasmic reticulum (ER), the Golgi apparatus and the nucleus. We used a *N. crassa* strain containing GFP-tagged NCA-1 protein, which is an ER marker protein, to observe the structure of ER [23]. A *N. crassa* strain containing GFP-tagged VRG-4 protein, which is a Golgi marker protein, was used to observe the structure of the Golgi apparatus [24]. DAPI staining was used to observe the structure of the nucleus. Using confocal microscopy, we examined the change in these organelles before and after HSAF treatment. The results demonstrated that the abundance of ER and Golgi apparatus dispersed in the hyphae were significantly decreased in HSAF-treated hyphae compared to the NO-HSAF treated control (Figure 6C,D), indicating that HSAF treatment may lead in part to a degradation of the ER and Golgi. For the nucleus, without HSAF treatment, multiple nuclei were observed to disperse in the whole mycelium, however, after HSAF treatment, no nuclei were observed in the mycelium (Figure 6E), suggesting that HSAF treatment resulted in the degradation of nuclei.

Based on the Atg8-family interacting motif analysis, 58 other co-occurring gene ontology terms were identified (Appendix A). It is noteworthy that a gene ontology term (GO:0044804) concerning nucleophagy was included in these ontology terms, implying a possibility that HSAF treatment could result in autophagy-mediated nucleus degradation in *N. crassa*. Together with confocal microscopy observation data, we speculated that HSAF treatment would cause autophagy, especially nucleophagy, and eventually result in cell death.

## 4. Discussion

Pathogenic fungi could cause disease in plants, animals and humans, and threaten food security, biodiversity and health [30]. However, due to the similar eukaryotic features between pathogenic fungi and their host, the number of antifungal drugs developed for the treatment of invasive fungal infections is quite limited compared to those available to treat bacterial infections. Currently, the target of commercially available antifungal drugs is mainly focused on a certain component of the cell membrane or cell wall of fungal cells, although new antifungal targets involved in other basic eukaryotic processes including fatty acid, ergosterol and ribosome biosynthesis have been investigated [31]. Considering that an antifungal drug must be an agent that selectively removes fungal pathogens from a host with minimal harmfulness to the host, elucidation of the mode of action of a potential antifungal compound is very important for their commercial application. It is clear that basic research on the discovery and the interaction mechanism of new antifungal drugs is needed.

The Gammaproteobacterial genus *L. enzymogenes* has obvious antagonistic effects on many microorganisms and has great potential application in agricultural, environmental and pharmaceutical industries [32]. HSAF, the secondary metabolite of *L. enzymogenes*, is known to have antifungal effects. Though great efforts have been made to investigate the mode of action of HSAF, the specific antifungal mechanism of HSAF has been still unclear due to the lack of molecular, genetic and biochemical techniques developed for most studied fungi. In this study, we used the model filamentous fungus *N. crassa* to investigate the antifungal mechanism of HSAF. Similar to the phenotype observed in *A. nidulans* [16] and *C. albicans* [18], HSAF significantly inhibited the germination of *N. crassa* conidia, as well as the aerial hyphae growth in a dose-dependent manner (Figure 1), suggesting that *N. crassa* would be a good model to study the antifungal mechanism of HSAF.

Comprehensive transcriptome analysis combined with microscopy observation showed that HSAF treatment led to increased expression of about 36 cell wall-related genes including seven chitin synthase genes and thicker cell walls (Figure 2). The hyper-accumulation of chitin in the cell wall is a common compensatory response to cell stress in many fungal species, such as *Kluyveromyces lactic* [33], *Aspergillus niger* [34], *Candida albicans* [35] and *Saccharomyces cerevisiae* [36]. In addition to *N. crassa* in our study, similar cell wall thickening caused by HSAF has also been observed in other fungal species including *Fusarium graminearum* [37], *A. nidulan* [16], *Bipolaris sorokiniana* and *Cryptococcus neoformans* [38], which indicates that enhanced cell wall synthesis is a general reaction to HSAF treatment for fungal species. It has been reported that amphotericin B (AmB)-tolerant *Candida tropicalis* isolates showed an enlarged cell wall and higher levels of β-1, 3-glucan in the cell wall, compared to the isolates of regular susceptibility [39]. In addition, higher β-D-glucan composition has also been found in AmB-resistant *Aspergillus flavus* isolates [40]. These reports suggested that enhanced cell wall synthesis was related to antifungal resistance, indicating that HSAF treatment could promote an antifungal resistance response in fungal cells.

Systematic mutants screening based on RNA-Seq analysis revealed that cell wall-related genes *chs-1*, *chs-9* and *gh16-5* played a negative role in the HSAF resistance, deletion of any of these genes led to an improvement in the HSAF resistance of *N. crassa* (Figure 4A), indicating that these genes could be related to the potential target of HSAF. Both *chs-1* and *chs-9* are chitin synthase encoding genes, and may be involved in cell wall synthesis. In theory, deletion of these genes should result in reduced HSAF resistance, which is contradictory to our results. Further investigation of the function of these genes showed that the lack of any of these genes did not affect the cell wall structure (Figure 4B), which could interpret our contradictory results shown in Figure 4A. Further SEM observation of the physical interaction between *L. enzymogenes* and *N. crassa* demonstrated that *L. enzymogenes* wildtype strains could adhere, aggregate and ultimately make pores in the surface of *N. crassa* wildtype strains (Figure 5). The HSAF no-producing *L. enzymogenes* mutants lost the ability to punch holes in the surface of any *N. crassa* strains, although these HSAF no-producing *L. enzymogenes* mutants could adhere and aggregate on the surface of *N. crassa* wildtype strains (Figure 5). On the other hand, *L. enzymogenes* wildtype strains could adhere and aggregate on the surface of any *N. crassa* strains, however, they were able to only punch holes in *N. crassa* wildtype strains (Figure 5). Moreover, HSAF no-producing *L. enzymogenes* mutants even lost the ability to aggregate on the surface of ∆*chs-1*, ∆*chs-9* and ∆*gh16-5 N. crassa* mutants. These data suggested that HSAF from bacteria, enzymes encoding by *chs-1*, *chs-9* and *gh16-5* genes from fungi are all required in the infection process, implying a possibility that the HSAF target may be related to fungal cell wall. The lack of a cell wall in mammalian cells and the different cell wall composition between fungal cells and plant cells make fungal cell walls an attractive target for development of antifungal agent. Although the interaction detail between HSAF and the fungal cell wall still needs to be investigated, our data so far supports that HSAF could be safe for humans or animals.

WGCNA analysis of the RNA-Seq data revealed that HSAF treatment caused upregulation of the genes associated with autophagy, especially nucleophagy, over time. Confocal microscopy observation of the structure of autophagy-related organelles ER and Golgi apparatus, as well as nuclei, further confirmed that autophagy was involved in the stress response to HSAF treatment (Figure 6). In response to devastating stimuli, living cells will undergo a genetic program known as programmed cell death (PCD) to ultimately lead to death. This suicide process can take several forms including apoptosis, necrosis and autophagy, each of which exhibits typical characteristics. These processes are usually interchangeable and sometimes difficult to distinguish [41]. Our data suggested that autophagy participate in HSAF caused cell death in *N. crassa.*

In conclusion, our results suggested that HSAF might interact with the fungal cell wall to assist *L. enzymogenes* in entering the fungal cells, thereby releasing more HSAF to cause autophagy-mediated degradation and eventually lead to fungal cell death. The findings of this study provided new perceptions about how *L. enzymogenes* interacted with its fungal prey and ultimately kill it.

## Figures and Tables

**Figure 1 jof-08-00252-f001:**
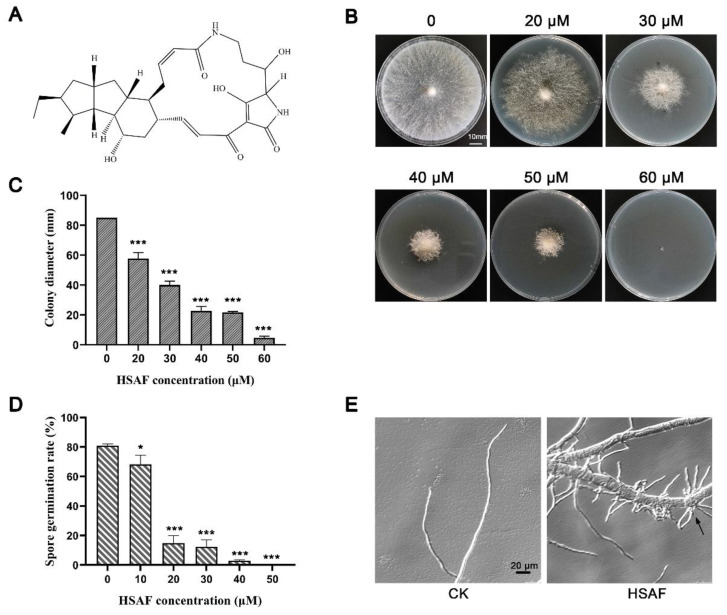
HSAF caused growth defect in *N. crassa*. (**A**) Chemical structure of HSAF; (**B**) Growth of *N. crassa* on Vogel’s plates supplemented with HSAF (0, 20, 30, 40, 50, 60 µM) for 36 h. Images show the results of one out of three experiments, and the statistical analysis of colony diameter is shown in (**C**); (**D**) Germination rate of *N. crassa* under HSAF treatment. Asterisks indicate significant differences (*, *p* < 0.05; ***, *p* < 0.001); (**E**) Microscopic morphology of *N. crassa* grown on Vogel’s media without (CK) or with 30 µM HSAF treatment (HSAF) for 6 h. The black arrow indicates the hyper-branch.

**Figure 2 jof-08-00252-f002:**
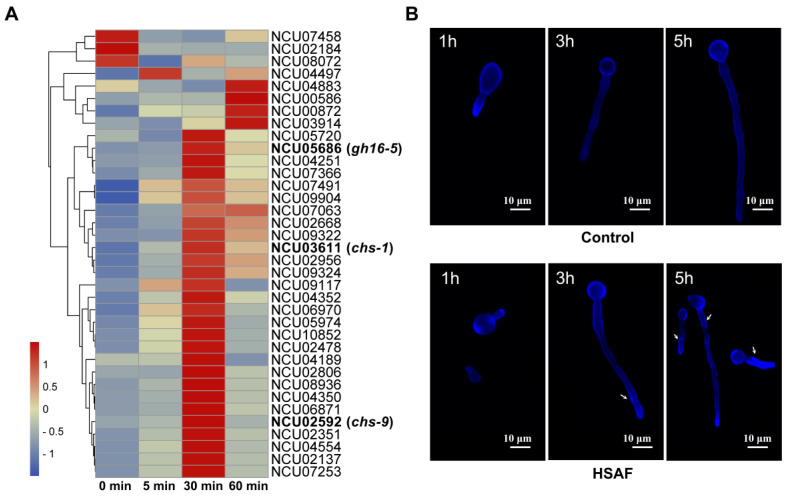
The effect of HSAF treatment on the formation and remodeling of cell wall. (**A**) The heat map of DEGs related to fungal-type cell wall organization. The heat map was constructed by R software pheatmap package with the original FPKM value of each gene; (**B**) Microscopic morphological characteristics and cell wall staining by CFW at different time of spore germination (1 h, 3 h, 5 h). The arrows indicate the regions with the thickened cell wall. Images show the results of one out of three replicates.

**Figure 3 jof-08-00252-f003:**
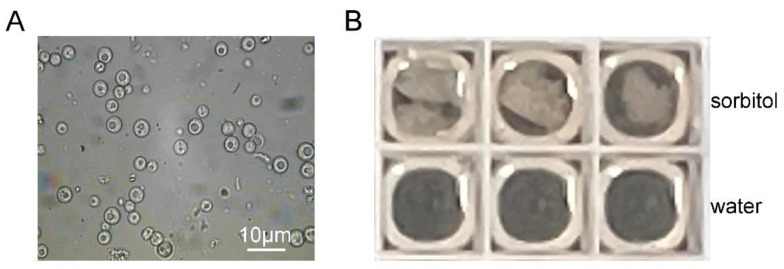
HSAF does not affect protoplast regeneration. (**A**) Protoplasting of *N. crassa*; (**B**) Protoplast regeneration tests with protoplasts suspended in sorbitol or distilled water. 10^6^/mL protoplasts were inoculated into Vogel’s media supplemented with 1 M sorbitol and 30 μM HSAF.

**Figure 4 jof-08-00252-f004:**
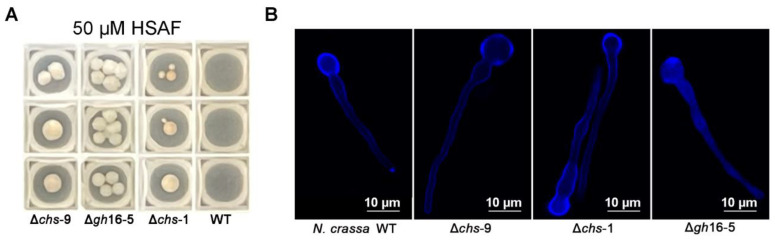
Deletion of gene *chs-9*, *gh16-5,* or *chs-1* in *N. crassa* increased the HSAF resistance. (**A**) Growth phenotype of the indicated mutants and wildtype of *N. crassa* on Vogel’s media with 50 μM HSAF; (**B**) CFW staining of cell wall of wildtype and the indicated mutants of *N. crassa*. Images show the results of one out of three replicates.

**Figure 5 jof-08-00252-f005:**
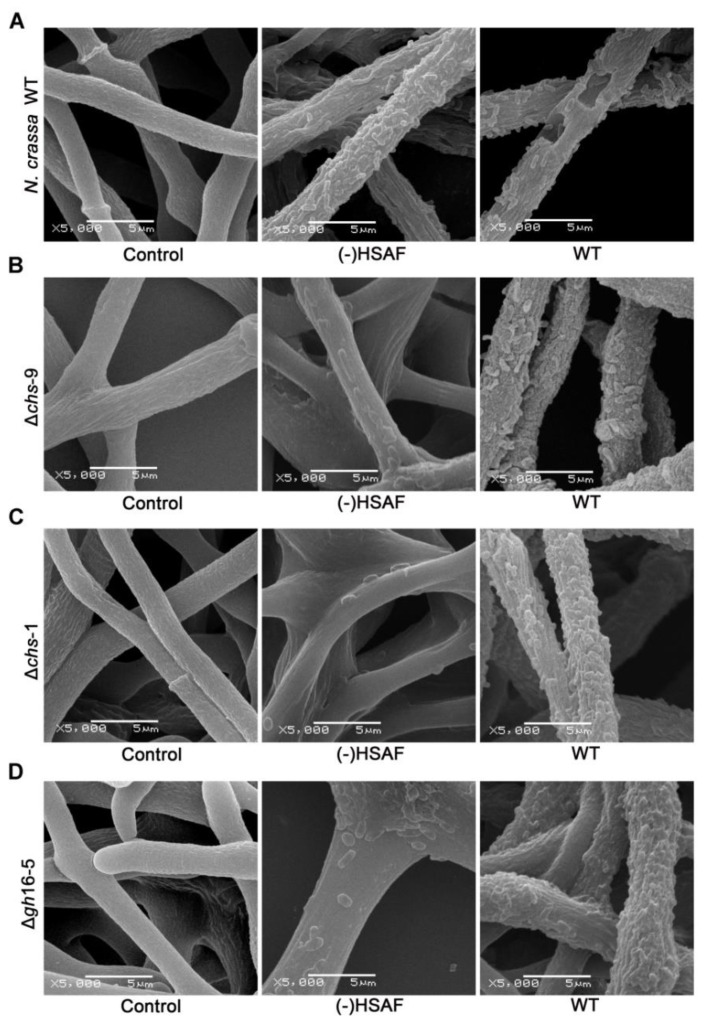
Physical interactions between different *L. enzymogenes* mutants and *N. crassa* mutants. (**A**) *N. crassa* WT; (**B**) *N. crassa* Δ*chs*-9 mutant; (**C**) *N. crassa* Δ*chs*-1 mutant; (**D**) *N. crassa* Δ*gh16-5* mutant interacted with *L. enzymogenes* strains. Control indicates growth of *N. crassa* alone, WT indicates *L. enzymogenes* wildtype strain and (-)HSAF indicates no-producing HSAF *L. enzymogenes* strain. Images show the results of one out of three replicates.

**Figure 6 jof-08-00252-f006:**
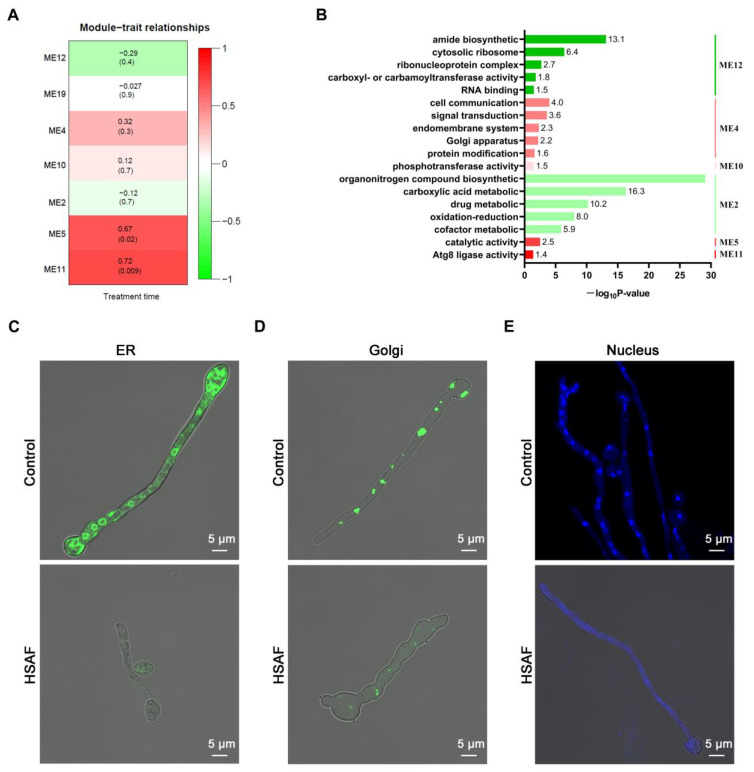
HSAF treatment may trigger autophagy-mediated degradation in *N. crassa*. (**A**) Correlation heat map of Modules and HSAF; (**B**) Functional category of DEGs in different modules; (**C**) Confocal microscopy of cellular morphology from germlings containing sGFP-tagged NCA-1 (ER marker); (**D**) Confocal microscopy of cellular morphology from germlings containing sGFP-tagged VRG-4 (Golgi marker); (**E**) Confocal microscopy of cellular morphology from germlings stained with DAPI. The germlings were grown in Vogel’s medium supplemented with 30 μM HSAF for 5 h. Images show the results of one out of three replicates.

## Data Availability

The RNA-seq raw data is available in NCBI’s Gene Expression Omnibus and are accessible through GEO Series GenBank accession NO: GSE195441.

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
