# Peer review of "Evaluating the Mode of Antifungal Action of Heat-Stable Antifungal Factor (HSAF) in *Neurospora crassa"

_jof, 2022, doi:10.3390/jof8030252_

Round 1

Reviewer 1 Report

The work presented for review, apart from a few editorial errors, does not raise any major objections.
I wish the authors success in their continued research.

Author Response

Response: We thank the reviewer for his/her support and positive comments on our study. We made a very careful proof-reading and all of the editorial errors have been fixed. Please see the track changes in the revised manuscript.

Reviewer 2 Report

the manuscript's English is correct and in general manuscript written well. However, authors did not explained their experimental methods in Materials and Methods sections. Statistical analyses section is not clear. There is no information  about how many replications? Also no information on negative or positive control. I think authors overused abbreviations which reduce readability. Please also see comments and corrections in the track changes.

Author Response

We thank the reviewer’s comments and suggestions. We noticed that all the above comments are included in the PDF file “peer-review-17721987.v1” with track changes. So we summarized each comment shown in this file and provided point to point response in the attached file.

Reviewer 2

   The manuscript's English is correct and in general manuscript written well. However, authors did not explain their experimental methods in Materials and Methods sections. Statistical analyses section is not clear. There is no information about how many replications? Also no information on negative or positive control. I think authors overused abbreviations which reduce readability. Please also see comments and corrections in the track changes.

Response: We thank the reviewer’s comments and suggestions. We noticed that all the above comments are included in the PDF file “peer-review-17721987.v1” with track changes. So we summarized each comment shown in this file and provided point to point response below.

  1. Line2: Title Are you "understanding the mode of antifungal action? Or Are you "Evaluating the mode of action ...?" In my opinion once you read the abstract you really evaluating the mode of action of the HSAF?? In general you do not use abbreviations in the heading evaluate

Response: We are appreciated for the reviewer’s comments. The title has been changed into “Evaluating the mode of Antifungal action of Heat-Stable Antifungal Factor (HSAF) in Neurospora crassa”

  1. Line15 significantly inhibit “How significantly?need a ‘%’

Response: We used different concentration of HSAF to test the effects of HASF on germination and aerial hyphae growth. The degree of the HSAF effect with different concentration was apparently different from each other and it is not necessary to show all of the quantitative data in the abstract. “Significantly inhibit” here is kind of descriptive result, rather than quantitative result. We thank the reviewer’s careful consideration, but we keep the original description based on the above explanation.

  1. Line19 WGCNA Spell out for the first time,Line54 AMPK Spell out for the first time

Response: We thank the reviewer for his/her careful proof-reading. The full name for WGCNA and AMPK were provided in the revised manuscript.

  1. Line 64 to 68 needs citation.

Response: We thank the reviewer’s suggestion. Reference 19, which was cited right above this part also includes the information from Line 64 to 68. To make the manuscript more clear, we changed the citation site of reference 19. We put it in Line 68.

  1. Line87 indicated concentrations What is the indicated concentrations? where they are??

Response: The indicated concentrations were shown in the result section (Fig.1C and Fig. 1D). The experiments process was exactly same for each concentration of HSAF. So we did not provide the detail concentration here. To make it more clear, we changed the sentence here into “For the HSAF sensitivity assay, the wildtype N. crassa conidia were harvested with distilled water and inoculated into Vogel’s media plates with indicated concentrations of HSAF as shown in Fig. 1.”.

  1. Line92 LB plates What is LB plates and how many you used?

Response: LB is the abbreviation of Luria broth, which is a nutrient-rich media commonly used to culture bacteria in the lab. We provided the full name of LB and clarified the number of LB plates used in the revised manuscript.

  1. Line103 The samples How many samples you had? And how many samples were in the experiment? Did you have any replications?

Response: We do have replications for each experiment. All the image data in this manuscript actually showed the results of one out of three replicates. We clarified this in the legends of figure 1B and figure 2B. However, we forgot to clarify it in the rest of figure legends. We are very appreciated for the reviewer’s careful proof reading. We provided the replication statement in the legends of all of the image figures in the revised manuscript. In addition, we added the statistics mark in the new Fig. 1C and Fig. 1D.

  1. Line108 specimens How many speciemens?

Response: 36 speciemens. We clarified it in the revised manuscript

  1. Line113 deep 24 well plates How many? See previous comment

Response: For our RNA-seq analysis, we set 4 time points and 3 replicates for each time point. We totally have 12 samples. One 24 well plate was enough. The RNA-seq raw data of these 12 samples is available now in NCBI’s Gene Expression Omnibus and are accessible through GEO Series GenBank accession NO: GSE195441. To make the manuscript more clear, we provided more detail information here in the revised manuscript, please see the track change.

  1. Line133 geminated Delete, insert "germinated"

Response: We thank the reviewer for his/her careful proof-reading. “geminated” has been changed into "germinated" in the revised manuscript.

  1. Line167 significantly inhibited What is the "%"

Response: This comment is similar to comment 2, please see our response to comment 2.

Reviewer 3 Report

The article by Liu et al, describes the mechanism of a novel anitfungal compound HSAF isolated from Lysobacter in filamentous fungus Neurospora crassa. The article is very interesting and well written. Here are my comments:

1)It will be very useful if the authors quantify the CFW fluorescence either from the microscopy images or do some Flow cytometry analysis.

2)The authors selected genes chs9, chs1, and gh16-5 for their analysis. It will be very useful if the authors consider highlighting the expression patterns of these genes in the heat map in fig 2.

3) Does the authors have any explanations as to why the gene expression peaks out at 30 mins HSAF exposure while goes down at 60 min exposure?

Author Response

Reviewer 3

The article by Liu et al, describes the mechanism of a novel anitfungal compound HSAF isolated from Lysobacter in filamentous fungus Neurospora crassa. The article is very interesting and well written. Here are my comments:

1) It will be very useful if the authors quantify the CFW fluorescence either from the microscopy images or do some Flow cytometry analysis.

Response: We thank the reviewer’s suggestion. It is difficult to quantify the CFW fluorescence using Flow cytometry analysis due to its filamentous structure. All the image data shown in this manuscript is the representation of at least three replicates. It is very easy to tell the difference from the current data. So we don’t think it is necessary to provide quantified data.

2) The authors selected genes chs9, chs1, and gh16-5 for their analysis. It will be very useful if the authors consider highlighting the expression patterns of these genes in the heat map in fig 2.

Response: We thank the reviewer’s suggestion. These three genes were highlighted in the heat map data shown in the new Fig 2 of the revised manuscript.

3) Does the authors have any explanations as to why the gene expression peaks out at 30 mins HSAF exposure while goes down at 60 min exposure?

Response: We do provide some explanation in the manuscript. Please see the text “The interaction between HSAF and its target will ruin cell wall integrity, which thereby trigger the cell to produce more cell wall components to temporally compensate for HSAF caused impairment.” in the end of section 3.2.

Reviewer 4 Report

It a very interesting  well written work that would find a wide interest in the antifungal research community. The topic is actual. Methods are appropriate  chosen and are in trend with the modern research.

Line 133 geminated change to germinated

Antifungal activity of HSFA was observed on the solid (agarized growth medium)  all other observation microscopy and RNA isolation was done after the cultivation in the liquid growth medium. Do the authors consider the mycelia grown on solid and liquid media as mycelia of the same quality? (Metabolism, physiology, aeration during the growth, gene expression...etc.)

Considering the selective toxicity of HSFA, was any toxicity research done on some mammalian cells? If the author have some information it would be important to include it into discussion.

Page 11 line 402  Do the author mean really  the resistance to antifungal agents or the tolerance that is a result of compensatory mechanisms incused after the treatment with antifungal active compound.

Page 12 line 422 These data suggested that HSAF from bacteria, products of enzymes encoded by chs-1, chs-9 and gh16-5 gene –I recommend the authors to include the words products of enzymes encoded by.. because genes are not directly involved in the interaction but  the structure that is a result of the enzyme activity (that is missing in the deletion strains).

Is somewhere in the literature available the information about the differences in the cell wall composition of these mutant strains and the wt? It would be helpful to understand the exact structure that is needed for the interaction of cell wall and the HSAF.

Page 12  line 428 ..our data so far supported that HSAF could be safe for humans or animals.. The toxicity results  on humans or animal cells are important for such statement –Authors have shown  another possible targets in the fungal cells  attributes of autophagy and nucleophagy   … these processes are important for animal cells and  the mechanism do not differ so much when compared with the fungal cells.

The authors had observed the autophagy in N.crassa as a result of stress caused by HSAF – Could the autophagy be not a signal for induction of some rescue mechanisms? It would be very interesting if the autophagy is observed in the mutant strains that show increased tolerance on HSAF , and if the autophagy mechanisms are observed after the treatment with in lower concentration of HSAF than the IC50 value?

Have the authors observed the cell wall pore formation after the treatment of the fungus with HSAF?

(I mean not with the bacterial cells only the fungus and the compound)

Author Response

Reviewer 4

It is a very interesting well written work that would find a wide interest in the antifungal research community. The topic is actual. Methods are appropriately chosen and are in trend with the modern research.

Response: We thank the reviewer for his/her support and positive comments on our study.

  1. Line 133 geminated change to germinated

Response: We thank the reviewer for his/her careful proof reading. “geminated” has been changed into "germinated" in the revised manuscript.

  1. Antifungal activity of HSFA was observed on the solid (agarized growth medium) all other observation microscopy and RNA isolation was done after the cultivation in the liquid growth medium. Do the authors consider the mycelia grown on solid and liquid media as mycelia of the same quality? (Metabolism, physiology, aeration during the growth, gene expression...etc.)

Response: We performed both solid and liquid media to test the effect of HSAF on its growth. HSAF could cause growth defect under both conditions. The data shown in Fig. 1E was performed under liquid media.

  1. Considering the selective toxicity of HSFA, was any toxicity research done on some mammalian cells? If the author have some information it would be important to include it into discussion.

Response: We thank the reviewer’s suggestion. But to our knowledge, there are currently no report about the effect of HSAF on mammalian cells.

  1. Page 11 line 402 Do the author mean really the resistance to antifungal agents or the tolerance that is a result of compensatory mechanisms incused after the treatment with antifungal active compound.

Response: We do not mean antifungal resistance resulted from the compensatory mechanisms incused after the treatment with antifungal active compound. Based on our data, HSAF treatment could cause thicken cell wall, which is a typical feature of AmB resistant fungi. So treatment with antifungal active compound could be a good trigger to generate antifungal resistance. But I don’t think it is the only way. I think any factors that can cause cell wall thickening are possible to result in antifungal resistance.

  1. Page 12 line 422 These data suggested that HSAF from bacteria, products of enzymes encoded by chs-1, chs-9 and gh16-5 gene –I recommend the authors to include the words products of enzymes encoded by.. because genes are not directly involved in the interaction but the structure that is a result of the enzyme activity (that is missing in the deletion strains).

Response: We are very appreciated for the reviewer’s careful proof reading. “chs-1, chs-9 and gh16-5 genes” has been changed into “enzymes encoding by chs-1, chs-9 and gh16-5 genes” in the revised manuscript.

  1. Is somewhere in the literature available the information about the differences in the cell wall composition of these mutant strains and the wt? It would be helpful to understand the exact structure that is needed for the interaction of cell wall and the HSAF.

Response: We checked almost all of the literatures related to these genes. We did not find any information about the cell wall structure. We thank the reviewer’s suggestion. It is really a good point to investigate the detail mechanism. We will perform this experiment in our next study.

  1. Page 12 line 428 ..our data so far supported that HSAF could be safe for humans or animals.. The toxicity results on humans or animal cells are important for such statement –Authors have shown another possible targets in the fungal cells attributes of autophagy and nucleophagy … these processes are important for animal cells and  the mechanism do not differ so much when compared with the fungal cells.

Response: Our study suggested that cell wall plays an important role in the interaction between HSAF and fungal cells. We have the solid proof that HSAF does not affect the regeneration of protoplasts (Fig. 3B). So we speculated that without cell wall, HSAF might not be able to cause autophagy mediated cell death. 

  1. The authors had observed the autophagy in N.crassa as a result of stress caused by HSAF – Could the autophagy be not a signal for induction of some rescue mechanisms? It would be very interesting if the autophagy is observed in the mutant strains that show increased tolerance on HSAF , and if the autophagy mechanisms are observed after the treatment with in lower concentration of HSAF than the IC50 value?

Response: From our RNA-seq data, the expression of autophagy related genes were significantly up-regulated in the last time-point (60 min treatment), while the expression of cell wall related genes were up-regulated in 30 min and down-regulated in 60 min, suggesting that autophagy could not be a signal for induction of some rescue mechanisms. It should be a signal for the process of cell death.

  1. Have the authors observed the cell wall pore formation after the treatment of the fungus with HSAF? (I mean not with the bacterial cells only the fungus and the compound).

Response: The cell wall pore was resulted from the lysis by cell wall degrading enzymes secreted from the bacteria. Fungal cell walls are unlikely to form holes without direct contact with the bacteria.

Round 2

Reviewer 2 Report

I would like to thank author/s providing extensive information and explaining their experimental method and replications and the statistical analysis. the manuscript reads much better with all corrections and additional information.